# Enzyme-substrate hybrid β-sheet controls geometry and water access to the γ-secretase active site

Shu-Yu Chen [1], Lukas P. Feilen [2], Lucía Chávez-Gutiérrez [3,4], Harald Steiner [2,5] & Martin Zacharias [1✉]

γ-Secretase is an aspartyl intramembrane protease that cleaves the amyloid precursor protein (APP) involved in Alzheimer's disease pathology and other transmembrane proteins. Substrate-bound structures reveal a stable hybrid β-sheet immediately following the substrate scissile bond consisting of β1 and β2 from the enzyme and β3 from the substrate. Molecular dynamics simulations and enhanced sampling simulations demonstrate that the hybrid β-sheet stability is strongly correlated with the formation of a stable cleavage-compatible active geometry and it also controls water access to the active site. The hybrid β-sheet is only stable for substrates with 3 or more C-terminal residues beyond the scissile bond. The simulation model allowed us to predict the effect of Pro and Phe mutations that weaken the formation of the hybrid β-sheet which were confirmed by experimental testing. Our study provides a direct explanation why γ-secretase preferentially cleaves APP in steps of 3 residues and how the hybrid β-sheet facilitates γ-secretase proteolysis.

[1] Center of Functional Protein Assemblies, Technical University of Munich, Garching, Germany. [2] German Center for Neurodegenerative Diseases (DZNE), Munich, Germany. [3] VIB-KU Leuven Center for Brain & Disease Research, Leuven, Belgium. [4] Department of Neurosciences, Leuven Research Institute for Neuroscience and Disease (LIND), KU Leuven, Leuven, Belgium. [5] Biomedical Center (BMC), Division of Metabolic Biochemistry, Faculty of Medicine, LMU Munich, Germany. ✉email: zacharias@tum.de

γ-Secretase is an aspartyl intramembrane protease complex that cleaves its substrates within the membrane bilayer. The catalytically active subunit is presenilin (PS, PS1, or PS2)[1–6]. Among several γ-secretase substrates, Notch1 and the amyloid precursor protein (APP) fragment C99 are particularly known for important biological functions as well as pathological processes. Notch1 serves as an essential factor in cell differentiation and several developmental pathways[7,8], while C99 is associated with Alzheimer's disease (AD)[9–11]. C99 contains 99 residues from the C-terminal fragment of APP and is generated upon cleavage of APP by β-secretase[12,13]. The remaining C99 fragment is subsequently cleaved by γ-secretase in a sequential manner within the lipid bilayer, producing the APP intracellular domain (AICD) and amyloid β (Aβ) peptides of different lengths[11]. It has been shown that D257 and D385 of PS1 located in transmembrane domains (TMD) 6 and 7, respectively, are essential for substrate cleavage[2]. The three major C99 cleavage products are Aβ40 (corresponding to the N-terminal 40 residues) that represents the most abundant Aβ form, as well as Aβ42, and Aβ38. Aβ40 and Aβ42 are produced from two different product lines. These start with the initial ε49 and ε48 cleavages, respectively, in the C-terminal TMD part followed by a series of successive cleavages towards the N-terminus of the TMD in a stepwise fashion, mostly 3-residue, interval[14–16]. It is widely accepted that Aβ42, which is highly prone to aggregation and forming fibrils that deposit as senile plaques in the brain[17], is causative for AD[9,10]. Consequently, γ-secretase inhibitors (GSIs) and modulators (GSMs) have been developed to reduce Aβ42 production as potential treatment for AD patients[18]. While the development of GSIs has been abandoned due to severe side effects in clinical trials, GSMs may provide safe means of drug targeting γ-secretase in AD[18].

Besides the catalytic PS subunit, the γ-secretase complex contains three additional subunits, nicastrin (NCT), anterior pharynx-defective 1 (APH-1, APH-1a, or APH-1b), and presenilin enhancer 2 (PEN-2)[19–21]. Following a stepwise assembly of these subunits and endoproteolysis of PS into an N-terminal (PS NTF) and a C-terminal (PS CTF) fragment, the complex attains its active form[2,22,23] The atomic structure of the complex with PS1 as catalytic subunit was successfully resolved by cryo-electron microscopy (cryo-EM), in 2015, without bound ligand or substrate[24]. A first cryo-EM structure of γ-secretase in complex with a GSI was reported in the same year and recently followed by additional cryo-EM structural analyses of γ-secretases bound to different GSIs at atomic resolution in 2021[25–27]. The cryo-EM structures of substrate-bound holo-forms of γ-secretase with either bound Notch1 or C83 substrates (the product of APP shedding by α-secretase cleavage) were released in early 2019[28,29]. However, solving the structures required to introduce a disulfide bond between substrate and PS1 to covalently capture the enzyme-substrate interaction and a catalytically inactive PS1 D385A mutant to prevent substrate cleavage. The substrate-bound structures revealed a hybrid β-sheet formed between PS1 and the substrate at the cytosolic side of the complex. This hybrid β-sheet is composed of three components, including Y288-S290 from PS1 NTF (β1), R377-L381 from PS1-CTF (β2), and the P1'-P4' amino acids of the substrate (β3; Fig. 1a). In addition, L432 next to the crucial PAL motif of PS1 CTF also forms a hydrogen bond with the P2' residue of the substrate (Fig. 1a). In the bound conformation, P1' and P3' are pointing to the intracellular side of the membrane and attach to the hydrophobic surface of the TMD6a (L268-E277) of PS1 while P2' is wrapped by the S2' pocket formed by residues of PS1 TMD7, TMD8, and TMD9 (Fig. 1a, b). Removal of β1, β2, or a mutation of the nearby L432 to proline (disrupting the hybrid β-sheet) essentially abolished substrate cleavage, indicating an indispensable role in γ-secretase proteolysis[28,29]. Hence, these experimental results on these PS1 mutations do not proof but provide already strong evidence for the essential role of the hybrid β-sheet with the substrate. The interaction between TMD6a of PS1 and ligand/substrate P1' and P3' was also observed in all currently resolved GSI-bound and substrate-bound cryo-EM structures[26,28,29]. In addition, the property of the S2' pocket that surrounds the substrate P2' residue, has been characterized by a mutagenesis study as being size-limited for aromatic amino acids[16]. Structural and sequence information of C99 P1-P4' at the ε49 cleavage pose are illustrated in Fig. 1a–c.

Despite of the structural and biochemical supports on the significance of the hybrid β-strand cluster, it is still unclear what role it plays in substrate cleavage and how mutations in these regions alter the enzyme activity. To answer these questions, we performed molecular dynamics (MD) simulations of substrate-bound γ-secretase complexes. In total, nine different substrates (WT and eight mutations) derived from C99 were chosen to study the influence of length of the β3 region and mutations therein. With the observed β3 dissociation event in the restraint-free simulations, umbrella sampling coupled with Hamiltonian replica exchange (HREUS) sampling technique was implemented to study the thermodynamic properties and structural details along the β3 association/dissociation pathway. Compared to the regular umbrella sampling (US) scheme[30] which samples along the reaction coordinates (RCs) in single direction, HREUS enables the systems to reversibly travel in multiple directions along the predefined RCs by allowing the neighboring replicas to exchange their Hamiltonians during parallel simulations[31,32]. By sampling the intermediate states between the associated and dissociated states, processes of association and dissociation of β3 are sampled in parallel.

In this study, we show that β3 in the substrate forms only a stable β-sheet with PS1 residues when at least three residues are present beyond the scissile bond and that it is essential for stabilizing the catalytic geometry, which is directly related to PS activity[33–36], by excluding excessive water molecules from the catalytic center. Furthermore, cleavage assays showed that the cleavage efficiency is significantly reduced when substrate P3' is mutated to a proline, whereas a much stronger, almost complete loss of cleavability was observed when proline is introduced at substrate position P2'. The reduced cleavability of C99M51P was explained by a significant reduction in binding affinity and the failure of closing the gap between the substrate and PS1 L432, leading to water leakage to the active site. A very similar effect was also observed in the case when Phe is introduced at the substrate P2' position. In contrast, when P1' is mutated to Phe, the interaction between β3 and the TMD6a of PS1 is enhanced, and the substrate scissile bond is better positioned at the catalytic site. Moreover, we found that the D385-protonated PS1 possesses a stronger β3 association energy but retains fewer water molecules in the catalytic center than protonated D257. When deprotonated, D385 attracts water molecules towards the conserved GxGD motif of the active site which is shared by all PS family members and shown to be critical for γ-secretase activity[4,37–39].

We suggest that the β-sheet formed between substrate and protease near the catalytic center provides a sufficiently rigid enzyme-substrate arrangement that is critical to position the substrate scissile bond and the enzyme active site in a conformation compatible for a hydrolysis reaction and might be a common feature also shared with other intramembrane proteases.

## Results

**At least three residues following the substrate scissile bond are required to form a stable β3 and a high fraction of optimal active site geometries.** Cleavage of C99 APP C-terminal fragment

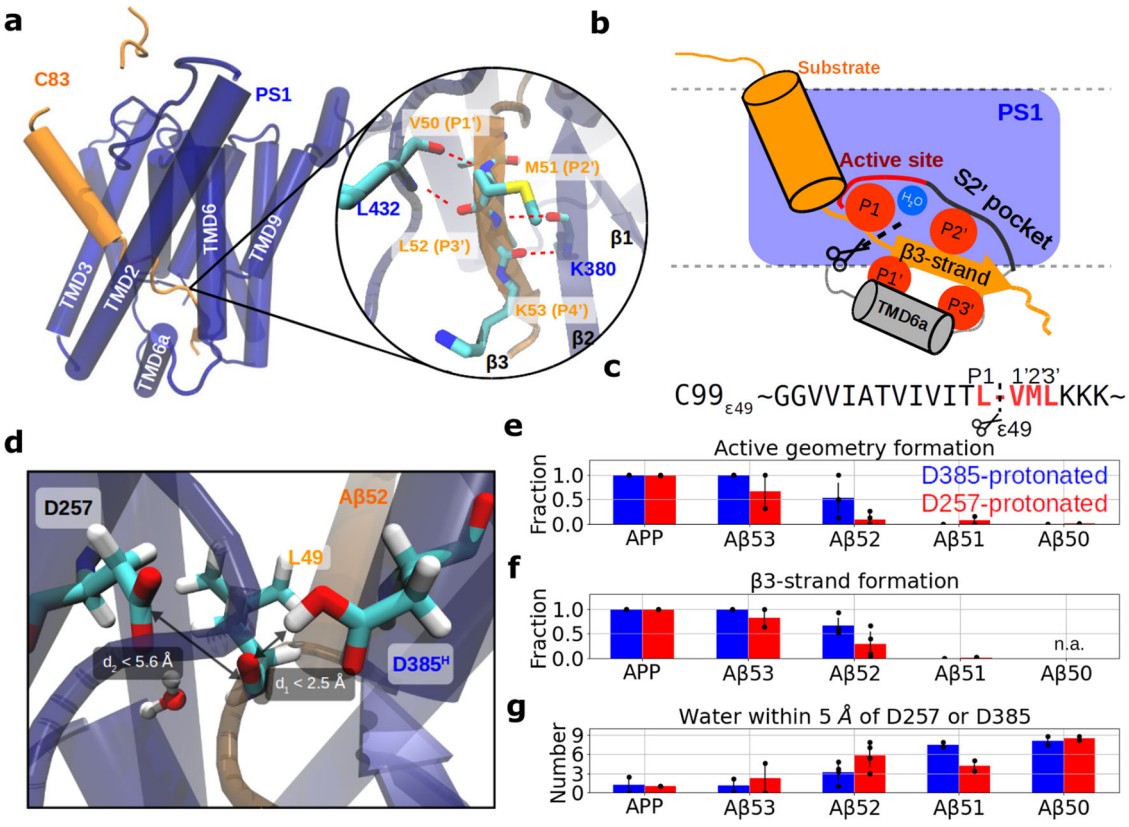

**Fig. 1 β-Strand interactions upon binding of APP and C-terminally truncated variants to presenilin/γ-secretase as studied by unrestrained MD simulations. a** Side view of presenilin (PS1) subunit (blue, TMD numbers in white) and bound APP substrate (orange) (cryo-EM structure PDB 6IYC). The interaction of the β1 and β2-strands (from PS1) with the β3-strand (from the APP substrate) is illustrated in the zoom-in panel. The backbone atoms of K380 and L432 of PS1, and the residues L49-L52 (P1–P3') of APP are shown as stick models and the hydrogen bonds between P2'-L432 and P3'-K380 are indicated by the red dotted lines). **b** Schematic view of the arrangement of PS1 (blue) and substrate (orange) at the active site. The scissile bond is located between substrate position P1 and P1' with P1 and a water molecule in the active site. P1' and P3' are in contact with TMD6a and P2' is accommodated in the size-limiting S2' subsite of PS1. **c** Sequence of APP C99 at the ε49 cleavage. P1 to P3' residues are colored in red with the cleavage bond denoted by the dashed line. **d** Illustration of a catalytically active geometry ready for cleavage formed by D257, D385$^H$, water, and substrate carbonyl group. It is characterized by an interval for $d_1$ being the distance of the catalytic hydrogen bond between the substrate scissile bond and the protonated aspartic acid and $d_2$ being the distance between substrate carbon and Cγ of the deprotonated aspartic acid. **e** Fraction of sampled states that form an active site geometry compatible with cleavage. **f** Fraction of sampled states that form a hydrogen-bonded β3-strand (hybrid β-sheet). Formation of hybrid β-sheet is not applicable in the Aβ51 substrate. (See "Methods" section) **g** Number of water molecules around the catalytic center in five different γ-secretase holo complexes (separate sets of simulations were performed for active site protonation states, color coded red (D257$^H$) or blue (D385$^H$). Time course data in **e–g** of individual simulations are shown in Supplementary Fig. S9. Error bars in **e–g** show the standard deviation of the mean of each system (n ≥ 2).

by γ-secretase proceeds consecutively in steps of typically 3 residues per cleavage event. Since the cleavage of C99 is relatively slow[40] and the soluble products may dissociate after any intermediate cleavage, one typically observes a series of Aβ peptides of different lengths originating from Aβ49 or Aβ48 in two major product lines starting from the initial cleavage at either the ε49 or ε48 cleavage sites[14]. Although it has been more than a decade since the successive tripeptide (and occasionally tetrapeptide) release from γ-secretase proteolysis was characterized[15,16], its molecular mechanism remains unclear. To investigate how the number of residues following the substrate scissile bond influences the active site dynamics and the substrate flexibility and cleavability, we generated several substrate-bound γ-secretase models in silico with the substrate being either APP or C-terminally shortened variants (see "Methods" section). The resulting shortened substrates are termed (in Aβ numbering) Aβ53, Aβ52, Aβ51, or Aβ50, by truncating APP after K53, L52, M51, and V50, respectively.

Structural data from X-ray and neutron diffraction show that one of the Asp residues in aspartate proteases is protonated and the second one is unprotonated[41–44]. However, no consensus has been reached on which Asp of PS1 is more likely to be protonated. Computational work with either D257 protonated[35,36,45], D385 protonated[33], or both unprotonated[46] in PS1 have so far been conducted.

Although D385 was suggested to be more prone to act as the general acid during the cleavage mechanism in the bound-state γ-secretase by a recent computational work with pH replica exchange molecular dynamics (pH-REMD) simulations[47], we are nonetheless interested in how PS1 protonation state changes the substrate binding and both possible PS1 protonation states, termed PS1-D257$^H$ and PS1-D385$^H$, are studied in all E-S complexes.

In total five models with wild-type (WT) substrate sequence were constructed and the dynamics were studied by multiple MD simulations in each case (at least 2 × 600 ns with each protonation state in PS1, see Table 1). Although the catalytically active geometry, i.e. the geometry of the active site compatible with substrate cleavage, in the following termed "active geometry", is often just defined by the hydrogen bond between the carbonyl group of the scissile bond and the protonated aspartic acid alone[48], such a geometry can also be catalytically incompatible

**Table 1 Simulation setup of γ-secretase in complex with different substrates.**

| Substrate | Sequence[a] | Simulation time (ns) | $N_{atoms}$ | Dimension (Å³)[b] |
|---|---|---|---|---|
| APP | ...**L₄₉**VMLKK... | 2 × 600 | 293,748 | 140 × 140 × 147 |
| Aβ53 | ...**L₄₉**VMLK | 2 × 600 | 293,639 | 140 × 139 × 139 |
| Aβ52 | ...**L₄₉**VML | 4 × 600 | 293,614 | 140 × 140 × 148 |
| Aβ51 | ...**L₄₉**VM | 2 × 600 | 293,595 | 141 × 140 × 147 |
| Aβ50 | ...**L₄₉**V | 2 × 600 | 293,578 | 140 × 140 × 148 |

The same set of simulations was performed for the two different PS1 active site protonation states.
[a]Sequence of each substrate is listed from standard cleavage position L49 on (marked bold).
[b]Box size measured using the last frame of the first simulation.

when the deprotonated aspartic acid partner is located far from the scissile bond and cannot serve as acceptor (Supplementary Fig. S1a, c). Hence, to better distinguish incompatible geometries from those that form an active geometry we define the latter one with two geometrical criteria exemplified in Fig. 1d and Supplementary Fig. S1a–d. Not only the carbonyl group of the scissile bond has to be hydrogen-bonded to the protonated aspartic acid ($d_1 < 2.5$ Å), but also the distance between the Cγ atom of the deprotonated aspartic acid and the substrate carbonyl carbon ($d_2$) has to be < 5.6 Å so that no more than one, at most two, water molecules can reside in the space in between, as otherwise the nucleophilic attack can be hardly executed. Snapshots of the hybrid β-sheet cluster and the active site of each E-S complex after 600 ns simulation time are shown in Supplementary Fig. S2.

As shown in Fig. 1e, we observed that the fraction of the active geometry formed in the simulations decreases for the C-terminally truncated substrate variants. Furthermore, simulations with PS1-D385[H] are statistically more prone to form an active geometry compared to those with D257[H]. In case of a WT substrate, the active geometry is found in over 99% of the sampled conformations in both protonation states. While the fraction decreases to 53% for the bound Aβ52 case with PS1-D385[H] and drops to 10% in case of PS1-D257[H]. For an even shorter substrate the active geometry at the catalytic center can barely be found. This result indicates a strong coupling between the number of residues C-terminal of the cleavage site and stability of the active site geometry ready for cleavage.

To better understand the molecular basis of the active geometry formation, we looked into the structural changes in the vicinity of the catalytic center. We identified two features, namely the fraction of the anti-parallel β-strand formed by residues M51 and L52 during simulations, indicated as the β3-strand formation fraction (see "Methods" section), and the number of water molecules surrounding the catalytic site, being most relevant for stable active site formation. Similar to the active geometry formation fraction, the fraction of the β3-strand decreases as the substrate gets shorter and is no longer stable in Aβ51. (Fig. 1f and Supplementary Fig. S2). When decomposing the β3-strand into the contributions from M51 and L52, we surprisingly found that L52 remains more stable in β-strand conformation although it is located closer to the substrate terminus, which is presumably more exposed to the aqueous environment and thus more accessible to water intervention. (Supplementary Fig. S3a, b) Secondary structures of the substrate TMDs over time are depicted in Supplementary Figs. S4 and S5 when binding to γ-secretase with PS1-D385[H] and PS1-D257[H], respectively. As the substrate is truncated to shorter peptides, the negatively charged COO⁻ substrate terminus approaches closer to the catalytic center and attracts more water molecules from the intracellular side (Supplementary Fig. S2). The water molecules brought from the C-terminus perturb the hydrogen-bonding network in the hybrid β-sheet and consequently access the catalytic center, suppressing the formation of an active

geometry. As shown in Fig. 1g, on average around one water molecule can be found within 5 Å of either D257 or D385 in the APP-bound and Aβ53-bound complexes. In contrast, more than three water molecules are found around at the catalytic center in the Aβ52-, Aβ51-, and Aβ50-bound complexes. Water is more prone to accumulate around the deprotonated (charged) aspartic acid than the protonated one (Supplementary Fig. S3c, d). Importantly, we observed that the water molecules around the catalytic site are trapped for a significantly longer period for APP and Aβ53 substrates, whereas shorter substrates like Aβ51 and Aβ50 usually result in a water residence times shorter than 10 ns (Supplementary Fig. S6). By looking into the correlation of these three features in each individual simulation, we identified that the formation of the active geometry correlates with the formation of β3-strand but anti-correlates with the amount of water molecules around the catalytic center (the β3-strand formation anti-correlates also with the amount of water molecules near the catalytic site, Supplementary Fig. S7).

In the cases of Aβ53 and Aβ52, where β3 is still formed, we observed that PS1 generally attracts more water around the catalytic center for the simulations with D257[H] than in simulations with D385[H] (Fig. 1g). The discrepancy in catalytic water accessibility might originate from the different micro-environments surrounding D257 and D385. Regardless of the choice of protonation state, V50 and L52 of the Aβ peptides contact the hydrophobic surface of TMD6a from PS1 (L268-R278), which we have recently proposed to be essential for substrate stabilization, creating a surface unfavorable for water access[34] (Supplementary Fig. S8a). In contrast, the GxGD motif (residues G382-D385 in PS1) which is highly conserved in the PS family, creates a small cavity compatible with water binding in the proximity of D385. When D385 is deprotonated, the negatively charged side chain attracts water molecules to the cavity near the GxGD motif (Supplementary Fig. S8b). Hence, mutating G382 and G384 in the GxGD motif into larger amino acids, which was found to virtually abolish γ-secretase activity[4,39], presumably blocks the water gateway. Notably, although PS1-D385[H] recruits less water molecules around the catalytic center, the water residence time in this protonation state is often found to be longer than in case of the PS1-D257[H] counterpart (Supplementary Fig. S6)

**Sampling the β3-strand association/dissociation by Hamiltonian replica exchange simulations.** Our multiple MD simulations with different substrates and protonation states indicate a strong coupling between the formation of an active geometry, the substrate β3-strand formation, and the water distribution around the active site. However, during the unrestrained MD simulations, a re-association of the substrate β3-strand with the β-strands of PS1 was rarely sampled (only few times in the Aβ52-bound complex, Supplementary Fig. S9) and it is therefore difficult to reach complete sampling convergence and to quantify the associated thermodynamic properties. To investigate the reversible β3 association and dissociation process, umbrella sampling (US)

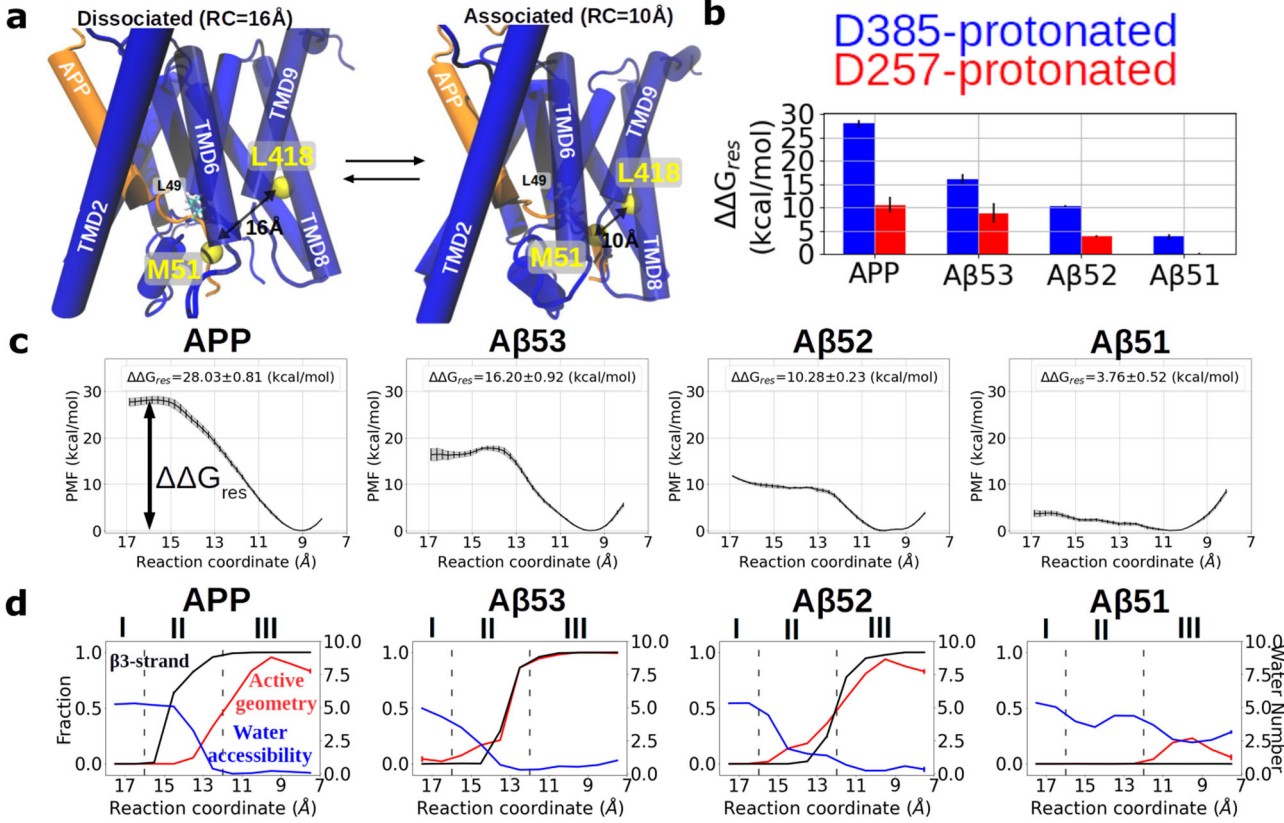

**Fig. 2 Hamiltonian Replica exchange MD along the β3-strand association pathway in APP and its truncated variants binding to γ-secretase-PS1-D385$^H$. a** Illustration of the dissociated state (substrate β3-strand dissociated from the L432 and β2-strand of PS1 with the reaction coordinate RC ≥ 16 Å) and the associated state (RC < 12 Å) of APP-bound γ-secretase. Cα atoms of PS1 L418 and substrate M51 are highlighted as yellow van der Waals beads. **b** The free energy difference between the associated state and dissociated state of substrate β3-strand. The protonation state of PS1 is color coded in red (D257$^H$) and blue (D385$^H$). **c** Potential-of-mean-force (PMF) profiles calculated with the HREUS method along the substrate β3-strand association reaction of APP and its truncated substrates. **d** Fraction of sampled states that form an active site geometry compatible with cleavage (red), β3-strand formation (black), and the number of water molecules around the catalytic center (blue), along the sampling pathway. Error bars in **b** and **c** show the standard deviation of the free energy in each HREUS simulation ($n = 16$). Error bars in **d** show the standard error of the features in each HREUS simulation ($n \geq 15$).

coupled with Hamiltonian replica exchange between the US windows (HREUS) was applied to the APP-, Aβ53-, Aβ52-, and Aβ51-bound complexes. It allows us to sample the intermediates along the β3-strand association and dissociation processes and to extract associated free energy changes. Among several potential reaction coordinates (RC), the Cα-Cα distance between PS1 L418 on TMD8 and substrate M51 was found to correlate well with the β3-strand dissociation event (Supplementary Fig. S10a, b) and provides a straightforward pathway of the β3-strand dissociation without severely distorting the rest of the substrate (Fig. 2a).

In the restraint-free simulations, this selected RC distance is below 10 Å when the β3-strand is formed and hydrogen bonds to the hybrid β-sheet with the PS1 β-strands. An RC distance above 13 Å indicates β3-strand dissociation from the PS1 β-strands (Supplementary Fig. S10b). To cover the entire β3-strand association and dissociation pathways, the chosen RC is sampled between 8 Å and 17 Å (see "Methods" section). However, the enforced dissociation/association along the RC without additional restraints resulted in a distortion of the enzyme β2-strand (in test simulations). To avoid such distortion, positional restraints were required on K380 of PS1, keeping PS1 in a ready-to-bind conformation, and on L418 in a stable geometry so that only the substrate strand is moving while the RC is increased. It is important to note that such restraint is expected to artificially stabilize the β3-strand associated state, but the bias is present in all cases and hence

should still allow us to obtain qualitative insight into the effect of shortening the C-terminus of the substrate. From the sampled configurations, the potential of mean force (PMF) profiles of γ-secretase complexes are calculated along the predefined RC using the weight histogram analysis method (WHAM[49]). The free energy difference required to bring the β3 away from the hybrid β-sheet region derived from the calculated PMF curve in the restrained system is denoted as $\Delta\Delta G_{res}$ (Fig. 2b, c). In addition, we also analyzed the fraction of sampled active geometry, β3-strand formation rate, and the amount of water near the catalytic center, denoted as water accessibility along the RC. For simplicity and clarity, we focus in the following on the binding processes with the PS1-D385$^H$ complex and results for the PS1-D257$^H$ complex are illustrated in the supplementary information (unless they deviate from the results for the PS1-D385$^H$ complexes).

Comparison on the WT and truncated substrates shows a clear trend of decreasing free energy differences between β3-associated (RC = 9–11 Å) and dissociated states (RC = 15–16 Å) in the order $\Delta\Delta G_{res,APP} > \Delta\Delta G_{res,A\beta53} > \Delta\Delta G_{res,A\beta52} > \Delta\Delta G_{res,A\beta51}$, regardless of the chosen PS1 protonation state (Fig. 2b and Supplementary Fig. S11). According to the change in the fraction of sampled active geometry and β3-strand formation rate, and the amount of water near the catalytic center, the conformations sampled along the RC can be roughly split into three regimes. These are the dissociated regime (regime I, RC > 16 Å), transition

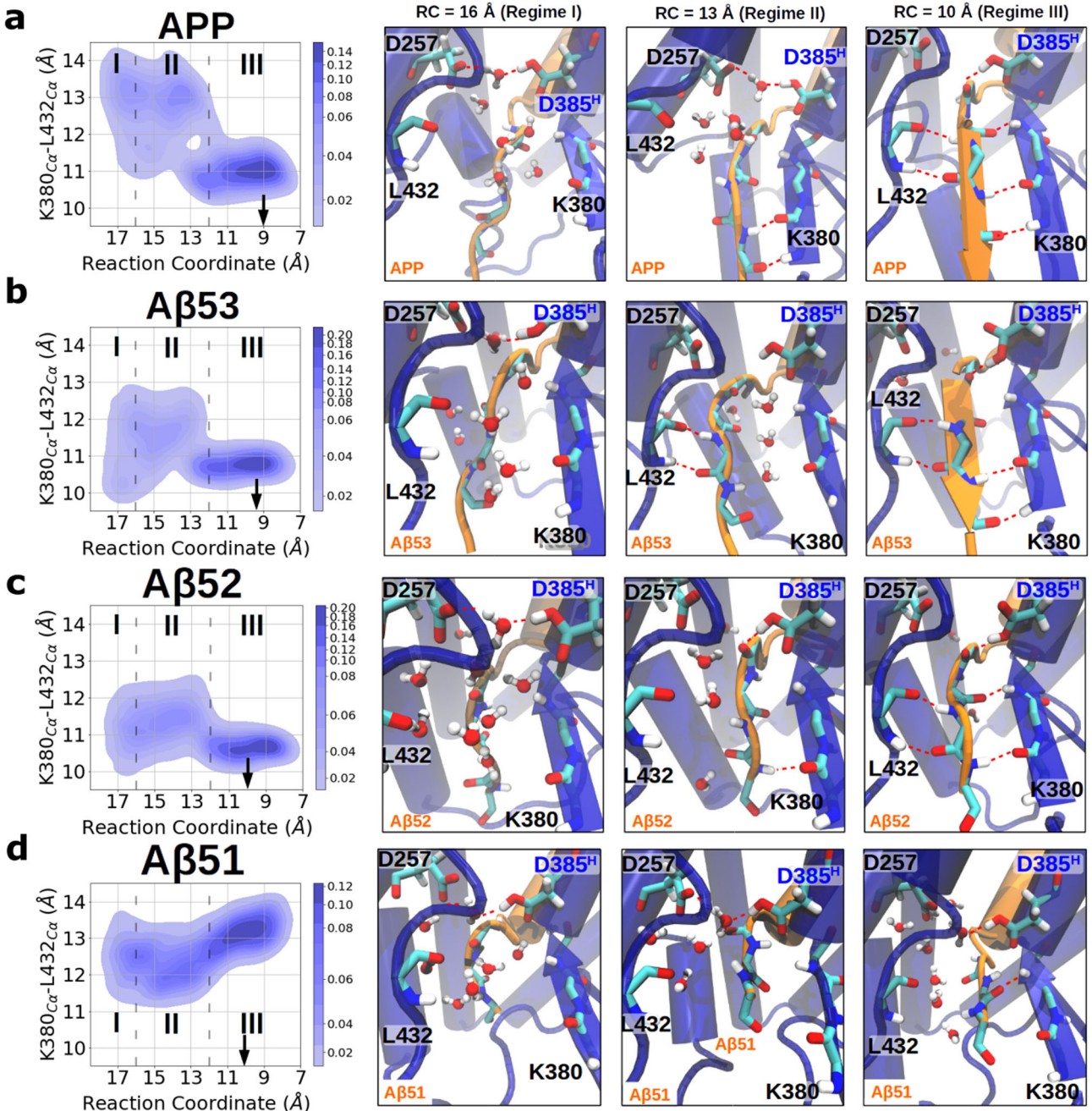

**Fig. 3 Gap width between L432 and β2 during the β3-strand association pathway in APP and its truncated variants binding to γ-secretase-PS1-D385[H].**
Distribution of the gap width between L432 and β2 along the association pathway and representative snapshots in regime I, regime II, and regime III when **a** APP, **b** Aβ53, **c** Aβ52, or **d** Aβ51 are bound to γ-secretase-PS1-D385[H]. The gap width between L432 and β2 is indicated by the Cα-Cα distance between L432 and K380. The black arrows point to the RC that corresponds to the PMF minimum. Three regimes are distinguished by the transparent dashed lines with regime I corresponding to the dissociated regime, regime II the transition regime, and regime III forming the associated regime. The substrates are shown in orange and γ-secretase is shown in blue. The coordinates of L49 to L52 (from up to down) of the substrates are shown additionally in the licorice representation.

regime (regime II, $12\,\text{Å} \leq \text{RC} \leq 16\,\text{Å}$), and associated regime (regime III, $\text{RC} < 12\,\text{Å}$) (Fig. 2d, representative snapshots of the E-S complex in each regime are shown in Fig. 3). In the dissociated regime (regime I), β3-association and the active geometry are barely formed and around five water molecules can be found at the active site periphery in all complexes (Fig. 2d). As the substrate is gradually approaching to form the hybrid β-sheet cluster (in regime II), β3-association and the active geometry begin to form, and neighboring water molecules are forced out of the active site region. In the associated regime (regime III), both the

active geometry and β3-strand are frequently formed with only few water molecules present around the catalytic site, except for the Aβ51 complex. A similar binding process is also observed in case of the PS1-D257[H] complex, however, in this case a few more water molecules are recruited into the catalytic center and the calculated binding affinity is weaker (Supplementary Fig. S11).

We next sought to understand how the formation of the β3-strand, active geometry, and water accessibility are influenced when M51 of the β3-strand is brought closer to the gap between β2 and L432 of PS1. We found that the hydrogen bond between L432 of

**Table 2 Cα-Cα distances between PS1 K380 and L432 in the available cryo-EM structures.**

| PDBID | Ligand | Hydrogen bond with PS1 | K385-L432 distance (Å) |
|---|---|---|---|
| 7Y5T | MRK-560 | Yes | 12.10 |
| 6LQG | Avagacestat | No | 12.11 |
| 6LR4 | Semagacestat | Yes | 10.68 |
| 7C9I | L-685,458 | Yes | 11.20 |
| 7D8X | L-685,458 & E2012 | Yes | 11.11 |
| 6IYC | C83 | Yes | 11.24 |
| 6IDF | Notch | Yes | 11.07 |
| 5FN2 | DAPT (not resolved) | n.a. | 8.42 |
| 5FN3 | Unknown helix | No | 5.09 |
| 5FN4 | Unknown helix | No | 11.13 |
| 5FN5 | None | n.a. | n.a. |
| 5A63 | None | n.a. | 11.27 |
| 4UIS | None | n.a. | n.a. |

Residue K380 is not resolved in PDBID 5FN5 and PDBID 4UIS.

PS1 and V50 of APP and the hydrogen bond between K380 of PS1 and M51 of APP might play important roles in this process. In the dissociated state (regime I), these two hydrogen bonds are not formed and the gap between β2 and L432 of PS1 is relatively flexible, indicated by the Cα-Cα distance between L432 and K380 fluctuating within 10 Å to 14 Å (Fig. 3a–d). This creates a gateway for water from the cytosolic side to access the catalytic center and perturbs the active site geometry. In this regime, a water bridging hydrogen bonding network between D257 and D385 was found in all E-S complexes, which is a thermodynamically favorable state of the active site geometry that we have previously described for the apo-form γ-secretase[33] (Fig. 3a–d).

In the transition state (regime II) M51 and L52 of APP can occasionally form a hydrogen bond with PS1 L432 and K380 on β2, respectively. The hydrogen bonds between the β3-strand and PS1 can efficiently block the water from accessing the catalytic center. When the β3-strand is brought to form a complete hybrid β-sheet cluster with PS1 (regime III), both hydrogen bonds are firmly formed and the Cα-Cα distance between L432 and K380 of PS1 is confined to ~11 Å (Fig. 3a–c), except for Aβ51 (Fig. 3d). The firm hydrogen bonds on both sides of the β3-strand prevent additional water molecules from the intracellular side to access the catalytic center and in turn strongly stabilize the active geometry (Fig. 3a–c). The closing of the gap between β2 and L432 of PS1 during the association process was also found in the PS1-D257[H] sampling (Supplementary Fig. S11c). Note that a Cα-Cα distance of around 11 Å between L432 and K380 was sampled as well in all resolved γ-secretase structures when GSI or substrate forms hydrogen bonds with K380 and L432 of PS1 (Table 2).

Interestingly, we observed that the initiation of the PMF gradient and the β3 association also correlate with the substrate length (Fig. 2c, d). While M51 and L52 start to turn into the β-strand conformation at RC > 15 Å in the APP substrate, this process started at RC ~ 14.5 Å and RC ~ 13.5 Å in Aβ53 and Aβ52, respectively (Fig. 2d). The earlier stages of PMF gradient and β3 association suggest that although the substrate residues C-terminal to L52 are not directly involved in β3-formation, they might indirectly facilitate the association process.

To understand the driving force of β3-association, we calculated the mean energy difference with respect to the dissociated state (RC = 16 Å), ΔΔH, contributed by the substrate V50-L52 segment along the RC. This was possible by using the Molecular mechanics Poisson-Boltzmann surface area method (MMPBSA, see "Methods"

section). Since the region of the hybrid β-sheet is located at the interface to the intracellular aqueous environment (distance to membrane lipids >15 Å) we did not use a MMPBSA approach with the protein emerged in a fixed implicit membrane but used the standard MMPBSA approach with a dielectric boundary of the protein region of interest and the nearby aqueous phase. We found that for all substrates except Aβ51, M51 contributes the most energy difference via Van der Waals (VDW) interaction, suggesting it plays a more important role than V50 and L52 in the β3 association event (Supplementary Fig. S12a, b). Notably, with the enthalpy gradient found in an early stage of the transition phase, M51 serves as the guide which brings the substrate V50-L52 together into the associated state when they are in the vicinity of PS1 β2 and L432. Meanwhile, we also observed that the residues at the substrate C-terminus, namely L52 of Aβ52 and M51 of Aβ51, contribute energy penalties due to electrostatic and polar solvation terms (EEL + POL; Supplementary Fig. S12b).

Taken together, the free energy calculations on γ-secretase in complex with APP and the truncated variants allow us to understand the functional role of the hybrid β-sheet and how substrate length and PS1 protonation state influence the active site geometry. Our result shows that the formation of a sufficiently stable hybrid β-sheet is essential for shaping of a stable active geometry and for limiting the amount of water molecules at the catalytic center by forming hydrogen bonds with L432 and K380 of PS1. In particular, the Aβ51 substrate does not form a stable β3-strand and coupled to this, also no active geometry at the catalytic site (Fig. 2d). Agreeing with the unrestrained MD simulations, HREUS simulations also indicate that PS1-D257[H] is more capable of water retention around the catalytic center while PS1-D385[H] binds the substrate β3 more strongly.

**APP M51P and L52P mutations decrease γ-secretase cleavage efficiency.** In the previous sections, we have demonstrated the essential role of a stable β3-strand in forming a catalytically competent active site geometry of γ-secretase. To further verify this hypothesis, we next sought to see whether specifically disrupting the formation of a stable hybrid β-sheet by substrate mutations may also influence γ-secretase cleavage. The amino acid proline can be used to disrupt hydrogen bonding in a β-sheet and we designed two APP substrate mutations M51P and L52P in silico. Similar to the free energy simulations on the bound wild-type (WT) substrate, the β3-association/dissociation was studied using the HREUS simulations in the APP$_{M51P}$- and APP$_{L52P}$-bound γ-secretase complexes (see "Methods" section). In comparison to the WT APP-bound complex, both substitutions lead to a decrease in ΔΔG$_{res}$ with the trend ΔΔG$_{res,APP}$ » ΔΔG$_{res,APP-L52P}$ > ΔΔG$_{res,APP-M51P}$ (Fig. 4a). In the APP$_{M51P}$-bound γ-secretase complex, both protonation states exhibit relatively flat PMF profiles (Fig. 4b and Supplementary Fig. S13a) compared to the WT APP substrate (Fig. 2c). In particular, the energy minimum no longer falls in the associated regime when APP$_{M51P}$ binds to PS1-D257[H] (Supplementary Fig. S13a). In contrast, the L52P mutant shows a qualitatively similar (but reduced) PMF profile compared to WT (Fig. 4b). Although the secondary structure of P51 and L52 in the APP$_{M51P}$ substrate can still stay in the β-strand conformations in the associated phase, the water molecules at the catalytic site are not effectively drained out as β3-strand is brought to form the hybrid β-sheet segment (Fig. 4c). Figure 4d shows that APP$_{M51P}$ fails to close the gap between L432 and β2 in the associated state (regime III). Despite hydrogen bond formation between L52 of APP$_{M51P}$ and K380 of PS1, the lack of a stable hydrogen bond between P51 of APP$_{M51P}$ and L432 of PS1 provides a water-accessible channel that consequently leads to a failure in forming a catalytically active geometry. In contrast, APP$_{L52P}$ is able to close the gap between L432 and β2

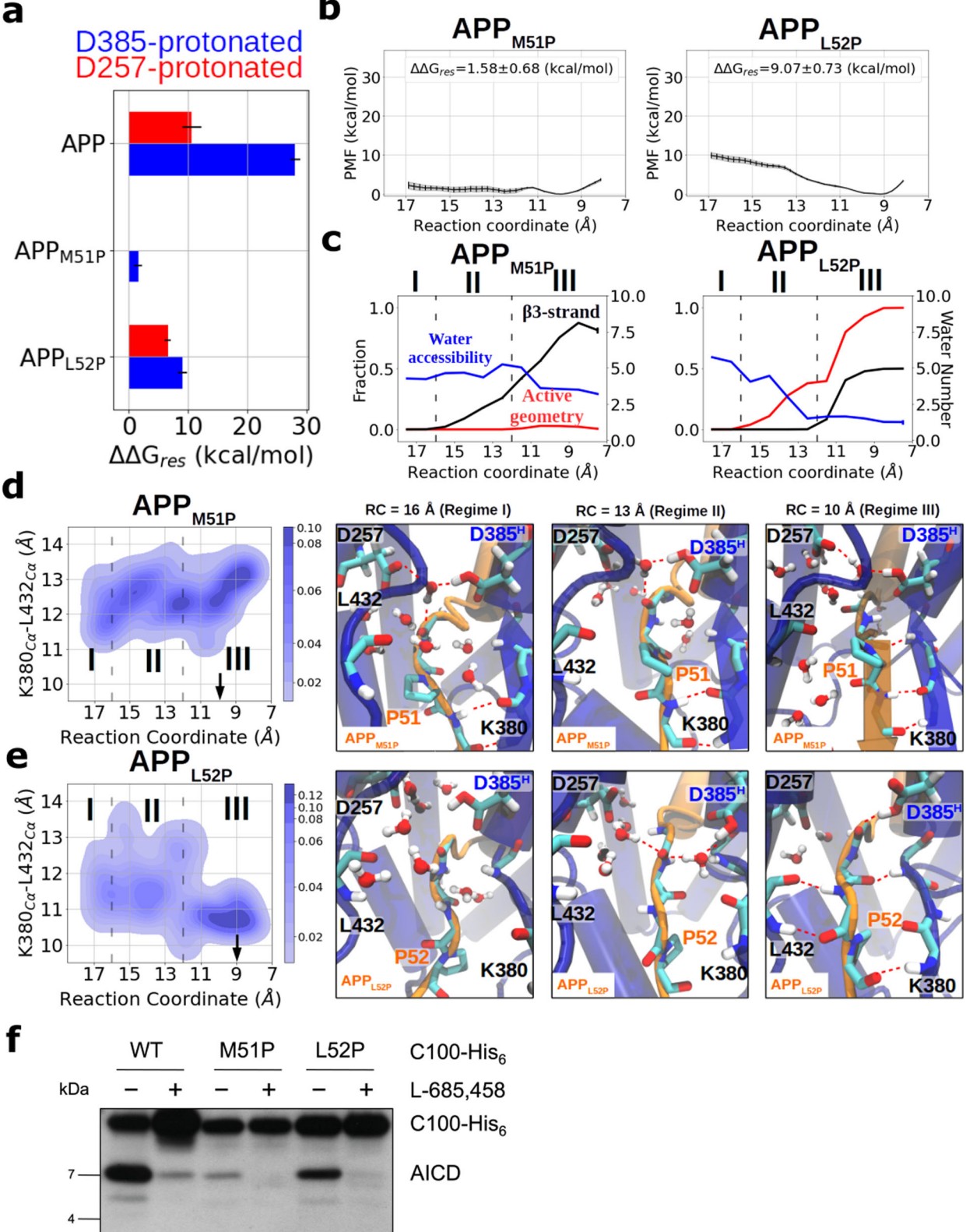

**Fig. 4 Effect of APP substrate mutations M51P and L52P on binding to γ-secretase-PS1-D385$^H$ and APP catalysis. a** The free energy difference between the bound state and unbound state of substrate β3-strand. The protonation state of PS1 is color coded in red (D257$^H$) and blue (D385$^H$). **b** PMF profiles calculated with the HREUS method along the substrate β3-strand association reaction coordinate of APP with mutation M51P or L52P. **c** Fraction of sampled states that form an active site geometry compatible with cleavage (red), β3-strand formation (black), and the number of water molecules around the catalytic center (blue), vs. the association reaction coordinate. **d–e** Change in the gap width between L432 and β2 along the sampling pathway and the representative snapshots when **d** APP$_{M51P}$ and **e** APP$_{L52P}$ are bound to γ-secretase. The atomic representations are shown in the same way as in Fig. 3. **f** Analysis of WT and mutant C100-His$_6$ cleavage by γ-secretase after overnight incubation at 37 °C by immunoblotting for AICD (Penta-His). Error bars in **a** and **b** show the standard deviation of the free energy in each HREUS simulation ($n = 16$). Error bars in **c** show the standard error of the features in each HREUS simulation ($n \geq 15$).

by forming a stable hydrogen bond with PS1 L432 and a weak hydrogen bond with K380 (Fig. 4e).

This prevents the water molecules from perturbing the active geometry and in the most thermodynamically favorable state (RC = 9 Å), the active geometry was found in more than 90% of the sampled configurations (Fig. 4C). Energy analysis shows that the ΔΔH gradient at P2′, the main interaction contributor in WT APP (Supplementary Fig. S12a), is significantly and mildly weakened in the M51P and L52P APP mutants, respectively (Supplementary Fig. S14a). Energy decomposition analysis shows that the M51P mutation attenuates the E-S interaction not only at position 51 but also mildly at positions 50 and 52 mainly in the VDW term (Supplementary Fig. S14b). L52P weakens the E-S interaction primarily by the loss in electrostatic interaction (EEL) on position 52 because of the lack of hydrogen bond with β2 and hydration penalty (POL) caused by increased water exposure at the substrate C-terminus (Supplementary Fig. S14b).

To prove our theoretical predictions, proline mutations were introduced at positions M51 and L52 of a C99-based recombinant C100-His$_6$ substrate[19] and their cleavability by γ-secretase was assessed by incubation with a purified γ-secretase complex composed of PS1, NCT, PEN-2 and APH-1a[50]. Indeed, compared to WT substrate the proline mutants were less efficiently cleaved by γ-secretase, as judged from the decreased levels of AICD (Fig. 4f). Remarkably, as predicted the M51P mutation very strongly inhibited γ-secretase cleavage, while the L52P mutation was much better tolerated. In accordance with previous data[51], L52P was exclusively cleaved after T48 (ε48; see MALDI-TOF mass spectrometry analysis in Supplementary Fig. S15) which can also be explained by our simulations. The P2′ position is more sensitive to the substitution compared to P3′. The cleavage site shift of the L52P mutant to ε48 places proline to the P4′ position, resulting in more favorable hybrid β-sheet formation. This explains why mutation L52P is efficiently cleaved at the ε48 site but M51P cleavage by γ-secretase is dramatically reduced (Supplementary Fig. S15). Importantly, a significant drop in cleavage efficiency was observed also in the shorter Aβ peptides when Pro is placed at the P2′ position such as T48P for the ζ46 cleavage[52]. This strongly suggests that the β3-PS1-hybrid β-sheet is also indispensable in the subsequent cleavage of shorter Aβ peptides.

**Simulations explain why Aβ-V50F enhances while Aβ-M51F mutation weakens the formation of the catalytically active geometry.** Since the major products from C99 during γ-secretase sequential trimming follow either the Aβ49-Aβ46-Aβ43-Aβ40 or the Aβ48-Aβ45-Aβ42 product line, the initial endoproteolytic cleavage at the ε49 or ε48 sites largely also controls the final products. Among many C99 mutations altering the final Aβ42/Aβ40 ratio, the size of the initial substrate P2′ position was suggested to have a direct impact on the ε-cleavage site since it needs to fit into a proposed size-limiting S2′ pocket in PS1, which is particularly problematic for bulky aromatic amino acids[16]. In contrast to the M51P and L52P mutations both the V50F and the M51F mutations in the substrate have already been studied experimentally very extensively and the cleavage behavior has been characterized. While ε49 cleavage is impaired for the C99 M51F mutant, ε48 cleavage is impaired in the V50F mutant. Similar results of M51F and V50F were also reproduced by direct AICD measurement[36] and can be deduced from the Aβ42/Aβ40 ratio[53,54]. Furthermore, the conventional production lines can be altered by introducing large Phe residues at other substrate positions, e.g. T48F and I45F, hypothetically the P2′ residue for the ζ46 and γ43 and cleavage, respectively[16,53,54].

In order to better mimic a general substrate-binding scenario for the subsequent cleavage poses, we used the Aβ52 system, where three residues are present beyond the scissile bond at the ε49 cleavage pose, as a model. Phe was introduced in silico in the Aβ52-bound γ-secretase complex at substrate positions P1′ or P2′ to generate Aβ52$_{V50F}$- and Aβ52$_{M51F}$-bound complexes, respectively. The β3-strand association processes of these two complexes were sampled using the HREUS method. While the free energy difference ΔΔGres was increased in the V50F mutant, it is significantly decreased in the M51F mutant (Fig. 5a). The PMF profile of β3 association of the V50F mutant is very similar to the WT and a flat free energy profile with the PMF minimum shifted toward dissociation is observed (Fig. 5b). In the Aβ52$_{V50F}$ association process, water exclusion and β-strand formation are more completely executed than for WT (Fig. 5c), and the gap between L432 and β2 is efficiently closed by the hydrogen bonds between PS1 and the substrate (Fig. 5d). In contrast, Aβ52$_{M51F}$ fails to prevent water molecules from reaching the catalytic center and the active geometry is barely formed (Fig. 5c). Although Aβ52$_{M51F}$ is more capable of closing the gap between L432 and β2 than the other two impaired substrates Aβ51 and APP$_{M51P}$ in the associated state, the gap is nonetheless not tight enough to drain out the water molecules (Fig. 5e). Structural comparison between WT Aβ52 and the M51F mutant in the associated state shows that PS1 L432 deviated by around 4 Å away from its original position to create enough space to accommodate substrate F51 (Supplementary Fig. S16), supporting the idea of the "size-limiting" PS1 S2′ pocket suggested by previous work[16]. The same effects of Aβ52$_{V50F}$ and Aβ52$_{M51F}$ were also found when forming the PS1-D257$^H$ γ-secretase complex (Supplementary Fig. S17).

MMPBSA calculation reveals that introducing Phe at position V50 enhances the VDW interaction at position 50 and an earlier initiation of M51-PS1 interaction, compared to the WT Aβ52 (Supplementary Figs. S12a, b and S18). On the contrary, both position 50 and 51 are considerably weakened when M51 is mutated to Phe, mainly because of the loss in VDW interaction (Supplementary Fig. S18).

In Table 3, we list the residues of PS1 with a minimal residue-residue distance lower than 5 Å from substrate P2′ in our simulations. We note that these contacting residues are fully consistent with the S2′ pocket described by the cryo-EM structures, inhibitor-bound model but the alternative S2′ pocket Bhattarai et al. suggested using the Gaussian accelerated molecular dynamics approach[36] was not sampled at all in our simulations. Taken together, our results offer explanations at molecular detail why substrates with a large Phe at P2′ position shift the ε-cleavage site positions. The large side chain at P2′ not only weakens the binding free energy of the hybrid β-sheet with PS1, but also expands the gap between L432 and β2 which results in a water leakage to the catalytic center and destabilizes the active geometry. Our computational work also shows that mutating P1′ can increase the binding affinity of the substrate β3-strand, forming a more stable hybrid β-sheet with PS1 and a more stable active geometry. These findings agree with and nicely explain the experimental result that the ε49/ε48 ratio is increased in the C99 V50F mutation and decreased in the M51F as well as how γ-secretase skips cleavages when Phe is placed at the P2′ position in other processive cleavage steps.

## Discussion

The question of why γ-secretase cleaves the C99 substrate in steps of at least three residues is still largely elusive. In our recent computational work, we have related the three-residue-wise cleavage with the putative translational movement of the substrate 3$_{10}$-helix at the PS1 internal docking site[48]. In this

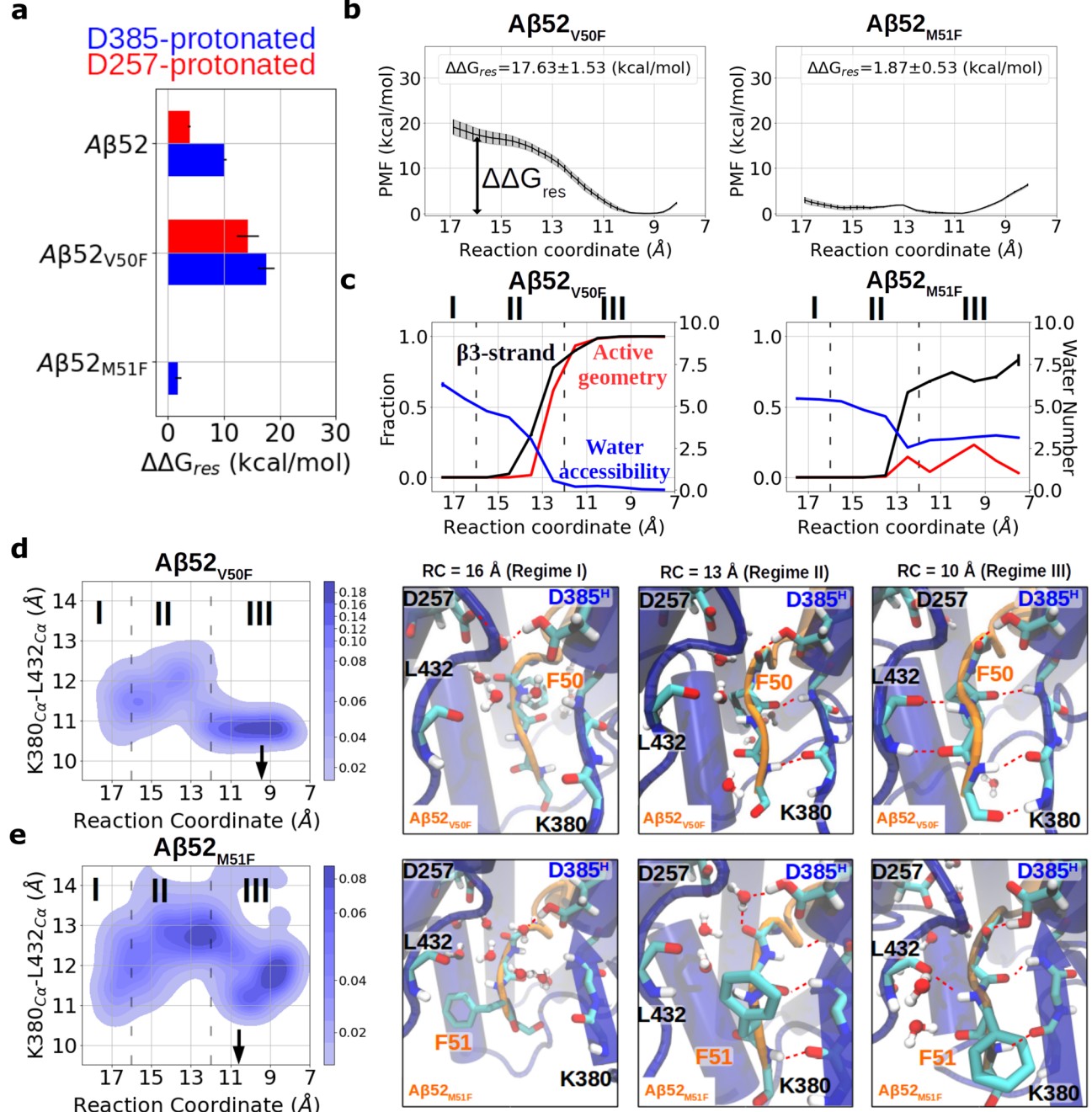

**Fig. 5 Free energy calculations and unrestrained simulations on V50F and M51F substitutions in the Aβ52 substrate bound to γ-secretase-PS1-D385ᴴ.**
**a** The free energy difference between the bound state and unbound state of substrate β3-strand. The protonation state of PS1 is color coded in red (D257ᴴ) and blue (D385ᴴ). **b** PMF profiles calculated with the HREUS method along the substrate β3-strand association reaction coordinate (RC) of Aβ52 with mutation V50F or M51F. **c** Fraction of sampled states that form an active site geometry compatible with cleavage (red), β3-strand (black), and the number of water molecules around the catalytic center (blue), vs. RC. **d**, **e** Change in the gap width between L432 and β2 along the sampling pathway and the representative snapshots when Aβ52 mutants (**d**) V50F and (**e**) M51F bind to γ-secretase. The atomic representations are shown in the same way as in Fig. 3. Error bars in **a** and **b** indicate the standard deviation of the free energy in each HREUS simulation ($n = 16$). Error bars in **c** show the standard error of the features in each HREUS simulation ($n \geq 15$).

study, we used a combination of molecular simulations and biochemical experiments to elucidate the role of the β-strand cluster and water distribution for forming catalytically active geometries at the active site of γ-secretase. Despite its stability being doubted by a computational work performed by Mehra et al.[46], the hybrid β-sheet C-terminal to the ε-cleavage sites, which consists of β1, β2, and β3, and PS1 L432 were shown to be critical for γ-secretase proteolysis[26,28]. Our simulations

indicate that β3 and consequently also the catalytically active geometry can only be stably formed when at least three substrate residues are present C-terminally of the scissile bond. The free energy profiles show for the longer substrates a strong β3 association affinity that diminishes for substrates with only 1 or 2 residues C-terminal of the scissile bond. Hence, this result offers a direct explanation of why γ-secretase cleaves APP in steps of three residues[28,29].

**Table 3 PS1 residues found in within 5 Å of any atom of substrate P2′ using the last frame of the window with RC = 9.8 Å in HREUS simulations.**

| Substrate | S2′ in PS1-D385$^H$ | S2′ in PS1-D385$^H$ |
|---|---|---|
| C99 | L85, V379, K380, L381, G382, L418, T421, L422, L425, A431, L432, P433, A434 | V82, L85, F86, V261, V379, K380, L381, G382, D385, L418, T421, L422, L425, K430, A431, L432, P433, A434 |
| Aβ53 | L85, V379, K380, L381, G382, D385, F386, Y389, L418, T421, L422, L425, A431, L432, P433, A434 | L85, V261, I287, V379, K380, L381, G382, D385, Y389, L418, T421, L422, L425, K430, A431, L432, P433, A434 |
| Aβ52 | L85, V261, V272, V379, K380, L381, D385, Y389, L418, T421, L422, L425, K430, A431, L432, P433, A434 | L85, V261, V379, K380, L381, G382, D385, L418, T421, L422, L425, K430, A431, L432, P433, A434 |
| Aβ51 | L85, V272, K380, L381, G382, D385, L418, T421, L422, L425, A431, L432, P433, A434 | L85, Q276, K380, L381, G382, D385, Y389, L418, T421, L422, L425, A431, L432, A434 |
| C99$_{M51P}$ | D257, V261, L268, V272, L381, G382, T421, L425, A431, L432, P433, A434 | V379, K380, L381, G382, D385, T421, L422, L425, A431, L432, P433, A434 |
| C99$_{L52P}$ | V261, V272, I287, V379, K380, L381, G382, L418, T421, L422, L425, K430, A431, L432, P433, A434 | V379, K380, L381, G382, D385, T421, L422, L425, A431, L432, P433, A434 |
| Aβ52$_{V50F}$ | L85, I287, V379, K380, L381, G382, L418, T421, L422, L425, A431, L432, P433, A434 | L85, V261, V379, K380, L381, G382, D385, L418, T421, L422, L425, K430, A431, L432, P433, A434 |
| Aβ52$_{M51F}$ | H81, L85, R377, V379, K380, L381, G382, L422, L425, L432, A434 | H81, L85, V379, K380, L381, G382, T421, L422, L425, A431, L432, P433, A434 |

Our simulations allow us also to draw a picture on the molecular details of the catalytic process: In the association process, the gap between β2 and L432 of PS1 serves as the bottleneck that recognizes and recruits the substrate P2′ residue into the PS1 S2′ sub-site, whereas β3 serves as a bottle plug which prevents redundant water molecules from flowing into the catalytic site. Before β3 association, more than five water molecules from the intracellular side can access the catalytic center through the gap between β2 and PS1 L432 and prevent hydrolysis by disturbing the topology at the catalytic center. Upon the association, β3 is attracted to the gap between β2 and PS1 L432 mediated by VDW interaction and drains the water from the catalytic center. In the associated regime, β3 zips up the gap between β2 and L432 by forming hydrogen bonds with them, and only less than two water molecules are trapped at the cleavage site allowing formation of a stable catalytically active geometry.

By assuming the binding poses of the ε48 cleavage is structurally similar to the ε49 cleavage, our simulation model is able to explain the shift in ε49/ε48 ratio observed experimentally, as illustrated in Fig. 6. In WT APP, M51 fits into the PS1 S2′ pocket and the hybrid β-sheet can efficiently control a limited water access to the catalytic center (Fig. 6a). Introducing a Pro to either the substrate P2′ or P3′ positions weakens the hybrid β3-sheet stability (Fig. 6b, c). Placing a Phe at the substrate P2′ position expands the gap between PS1 K380 and L432 because of the limited size of the PS1 S2′ pocket (Fig. 6b, c and Supplementary Fig. S19). In the M51P mutant, the hydrogen bond between substrate P2′ and PS1 L432 is lost, and this creates a water-accessible gateway. In the case of the L52P mutant, the substrate β3-strand is shortened, and the binding affinity is weaker than for the WT case, nevertheless, the simulations indicate that the active geometry was still sampled when the hybrid β-sheet is formed. However, the problem of weak β3-strand binding affinity of the L52P mutant can be circumvented by adopting the binding pose for the ε48 cleavage (Fig. 6c). Indeed, consistent with our model, ε48 cleavage in the L52P mutant was observed in our cell-free assay as sole ε-cleavage event. Exclusive or strongly preferred cleavage, respectively, at T48 has also been observed in previous experimental studies[52,55]. Moreover, also consistent with our model, the M51P substrate mutation caused a very strong drop in total activity in our assay.

In contrast to the Pro mutations, the M51F mutant does not alter the β3-strand stability but expands the gap between substrate and PS1 residue L432 and fails to block the water from

accessing the catalytic center when associated in the ε49 cleavage pose although energetically unfavorable, leading to a decrease in the ε49/ε48 ratio[16,36] (Fig. 6e). Along these lines, the V50F mutation should also suffer from water leakage through the same mechanism and should fail to form a stable active geometry when binding in the ε48 cleavage pose, hence, it should lead to an increase in the ε49/ε48 ratio which has been observed experimentally ratio[16] (Fig. 6d). It is important to add that our simulation-derived model can also explain the observed effect of Pro and Phe substitutions on several subsequent APP substrate cleavages. This includes the inhibition of the ζ46 cleavage (Aβ49 → Aβ46) by T48F[35] and T48P[52], the observed inhibition of the γ43 cleavage (Aβ46 → Aβ43) by I45F[16,52], and the inhibition of γ40 cleavage (Aβ43 → Aβ40) by A42F[16] (Supplementary Fig. S20), suggesting that the hybrid β-sheet conformation, especially at the P2′ position of the substrate, is also indispensable for the cleavage of shorter Aβ peptides.

Our study also gives important insights into the influence of the protonation states of the active site residues D257 and D385 in PS1. While qualitatively similar results were obtained for both protonation states in support of the role of the hybrid β-sheet, differences in active geometry, water recruitment and β3 association strength were observed. While the hydrophobic surface of TMD6a in contact with V50 and L52 of C99 makes the proximity of D257 more hydrophobic, the GxGD motif in TMD7 allows water to dwell in the cavity in the vicinity of D385. In the simulations, the D257-protonated PS1 on average recruited more water molecules to the catalytic center than the D385-protonated state. Meanwhile, in all the studied E-S complexes, D385-protonated PS1 binds substrate β3 more firmly at the β-sheet region and traps the limited water molecules longer around the catalytic center. Taking into account that the peptide hydrolysis results in the net proton transfer from the protonated aspartate to the unprotonated, the functional discrepancies between two protonation states might suggest that γ-secretase captures the substrate in the D385$^H$ protonation state and releases the product when the proton is transferred to D257. This could be subject of a future simulation study which allows sampling of different protonation states during MD simulations.

Our work provides mechanistic insight in how the hybrid β-sheet facilitates γ-secretase proteolysis by excluding water from the catalytic site and the fundamental reason of three-residue-wise cleavage on C99. Notably, an enzyme-substrate β-sheet and the size-dependent modulation near the catalytic center are also

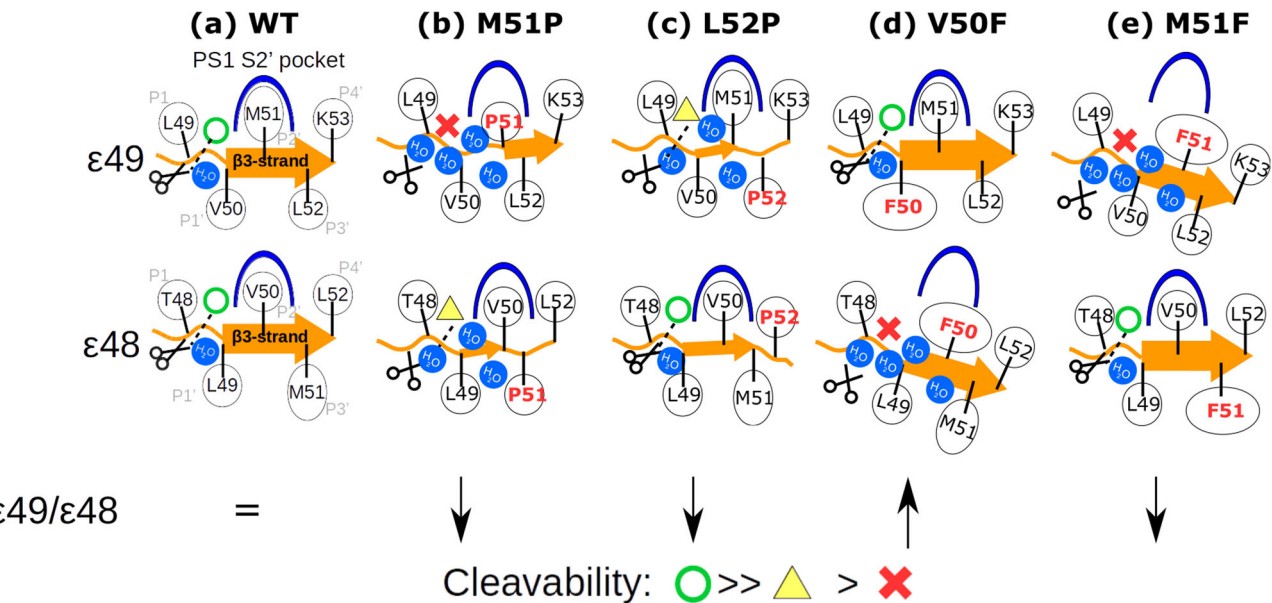

**Fig. 6 Schematics of the APP mutation effect on ε49/ε48 cleavage ratio according to our simulation model.** The ε49 cleavability of **a** WT, **b** M51P, **c** L52P, **d** V50F, and **e** M51F APP is illustrated according to our computational results. The cleavability at the ε48 cleavage site is estimated by assuming the same binding pose as in the ε49 cleavage with a one residue shift. The length of the β3-strand is monitored from the HREUS simulation (Supplementary Fig. S19) with the orange arrow and its thickness indicating the binding affinity to PS1. The hypothetical cleavability of substrates at either the ε49 or ε48 sites is indicated by a green circle (unaffected), yellow triangle (weakened), or red cross (inhibited). The PS1 S2′ pocket is colored blue and APP is represented in orange. The γ-secretase catalysis is represented by black scissors and water molecules are illustrated by blue circles. The ε49/ε48 ratios shown in the bottom panel are predicted according to our model.

found in rhomboid[56], an intramembrane serine protease, and the other wide range of proteases[57–61] (Supplementary Fig. S21). Hence, our study may offer not only insight into the E-S interaction in γ-secretase but could also be of general importance for a better understanding of intramembrane proteolysis.

## Materials and methods

**Simulation setup.** All substrate-bound γ-secretase complexes were initiated from the cryo-EM determined structure PDB 6IYC[28] with mutations D385A and Q112C of PS1 restored via Ambertools18 program[62]. By assuming that the TMD of APP-C99 binds to PS1 in a similar way as C83, atomic coordinates of C83 are taken for the APP-bound model. The five unresolved residues D23-N27 at the N-terminal side of APP and Q56-S59 at the C-terminal side of APP were modeled using the comparative modeling program MODELLER[63]. The other missing residues of C99 at the C-terminal and N-terminal sites were not reconstructed because of the uncertainties about their coordinates. In total, L17-S59 (Aβ numbering) of APP are represented in our APP-bound model. The APP-bound γ-secretase is first oriented via the OPM[64] server and embedded into a POPC membrane environment via the CHARMM-GUI[65] online server. In total 503 POPC and 68102 water molecules are added into the simulation box salted with 0.15 M KCl. The APP-bound γ-secretase complex was taken as a reference for all other substrate-bound systems to reduce any structural bias emerging from rebuilding the simulation box. Complexes with different substrate length are generated by deleting the unwanted residues at the C-terminal side and neutralizing the system by replacing water molecules to potassium ions via Ambertools18. Complexes with mutated substrates are constructed using Ambertools18. With both PS1-D385[H] and PS1-D257[H] being investigated, in total 18 systems are constructed for simulations and listed in Table 1. Several unrestrained 600 ns trajectories are generated in different systems according to how diverse the observed quantities are. All simulations were performed using the CUDA accelerated version of Amber18 PMEMD program[66]. Atomic interactions were described by the ff14SB[67] force field for the proteins, the TIP3P[68] for the water, and lipid17[69] for the POPC molecules.

All simulation systems were first energy minimized (70,000 steps) before the equilibration and production run. During an equilibration phase (0.6 ns), the motions of protein and POPC molecules were restricted using positional restraints with a force constant of 10 kcal mol⁻¹ Å⁻² on protein and 5 kcal mol⁻¹ Å⁻² on lipid with a temperature of 303.15 K. Subsequently, the positional restraints were gradually removed during the 0.6 ns simulation time. During data gathering simulations, the temperature was maintained at 303.15 K via a Langevin thermostat[70] and the pressure was kept at 1 bar via a Berendsen barostat[71]. The SHAKE algorithm[72] was applied for all bonds containing hydrogen. NPT

trajectories were generated for analysis with a time step of 4 fs using hydrogen mass repartitioning[73].

**β3 association/dissociation sampling with the HREUS method.** To sample the β3 association pathway, umbrella sampling (US) coupled with Hamiltonian replica exchange MD (HREUS) was implemented. To sample a reasonable association/dissociation pathway, the chosen reaction coordinate (RC) should sample the similar configuration states as what was sampled in the restraint-free simulations. With the knowledge of dissociated β3 state sampled in the restraint-free simulations, the Cα-Cα distance of PS1 L418 and substrate P2′ are chosen as the reaction coordinate to capture the dissociation and re-association path. With the low RMSF observed in L418 in restraint-free APP-bound simulations (Supplementary Fig. S22), it is reasonable to introduce positional restraint of 10 kcal mol⁻¹ Å⁻² on the L418 Cα atom to make sure that the measured RC relies totally on the movement of substrate P2′. In addition, positional restraint of 10 kcal mol⁻¹ Å⁻² was applied on PS1 K380 to avoid the β2 distortion observed in the dissociated simulations. Following the energy minimization and equilibration protocol used in the restraint-free simulations, each system was first simulated for 10 ns with harmonic distance restraint forcing the formation of the catalytic hydrogen bond, namely $d_1$, at 2.2 Å depicted in Supplementary Fig. S1 and the Cα-Cα distance between PS1 K380 and L432 was kept at 12 Å to avoid β3 dissociation at constant temperature of 303.15 K and constant pressure of 1 bar. The structure is then sequentially submitted to regular US protocol ranging from RC = 8.0 Å to RC = 17.0 Å with an interval of 0.6 Å to prepare the starting structure for the following HREUS sampling. The structure is sampled for 3 ns in each window, in total 16 windows are prepared with 48 ns NVT simulations. A force constant $k = 6$ kcal mol⁻¹ Å⁻² was applied on RC = 8.0, 8.6, 9.2, 9.8, 16.4, and 17.0 Å and a force constant $k = 8$ kcal mol⁻¹ Å⁻² was applied on RC = 10.4, 11.0, 11.6, 14.6, 15.2, 15.8 Å. A stronger force constant $k = 10$ k cal mol⁻¹ Å⁻² was applied on RC = 12.2, 12.8, 13.4, 14.0 Å to sample the processes around the higher free energy barrier. The generated structures in different RC windows were together submitted to the HREUS protocol using pmemd.cuda.mpi version of Amber18 program. Exchanges between the neighboring replica were attempted for every 10 ps with an exchange acceptance ratio between replica i, with a system configuration $\mathbf{r}^i$ and Hamiltonian $H^i(\mathbf{r}_i)$, and replica j, denoted as $P_{acc}(\mathbf{r}_i \rightarrow \mathbf{r}_j)$, follows the metropolis criterion

$$P_{acc}(\mathbf{r}_i \rightarrow \mathbf{r}_j) = \min[1, \exp(-\Delta)]$$
$$\Delta = \beta((H^i(\mathbf{r}_j)+H^j(\mathbf{r}_i)) - (H^j(\mathbf{r}_j)+H^i(\mathbf{r}_i)))$$
(1)

where Δ is the change in total energy upon the replica exchange attempt and $\beta = 1/(k_B T)$ is the reduced inversed temperature with Boltzmann constant $k_B$ and the physical temperature T. In total, 60 ns of NVT sampling were carried out in 16 replicas with the same force constants and RC spacing used in the US protocol. The

first 20 ns of the HREUS trajectory was considered as the equilibrium process between the windows and only the later 40 ns were taken for the following analysis. The potential of mean force (PMF) profiles were calculated with the weighted histogram analysis method (WHAM[49]). The sampled RC and convergence of PMF are shown in Supplementary Fig. S23 and Supplementary Fig. S24, respectively. $\Delta\Delta G_{res}$ is taken from the PMF value at RC = 16 Å to describe the free energy difference between the associated and dissociated state if the global PMF minimal falls in the regime with RC < 11 Å and zero otherwise.

**Characterization of active site geometry and extraction of energy contributions.** To better illustrate the quantities related to the geometry at the active site, fraction of active site geometry and β3 formation, and the amount water molecules around the catalytic site are measured in both restraint-free and HREUS simulations. A geometry is only considered as a catalytically active one when a catalytic hydrogen bond is formed between the scissile carbonyl and the protonated aspartic acid (depicted as $d_1 < 2.5$ Å in Supplementary Fig. S1) and the distance between the unprotonated aspartic acid is not too far away from the scissile carbonyl so that the proton transfer is still achievable (depicted as $d_2 < 5.6$ Å in Supplementary Fig. S1). During the simulations of the APP-bound complex, P1–P4', namely V50-K53, occasionally turned into an anti-parallel β-strand conformation. (Supplementary Fig. S4) Compared to substrate P1' and P4', P2' and P3' formed more stable hydrogen bonds with PS1 and stayed more steadily in the β-strand conformation in the APP-bound complex. The completeness of β3 is thus computed by taking the average β-strand occupation fraction of P2' and P3' calculated by the DSSP method[74], namely β3 = [β(P2') + β(P3')]/2. Since Aβ51 does not have P3' residue, only the β-strand occupation fraction of P3' is taken into account, namely β3 = β(M51). A water molecule is considered being around the catalytic center if any atom of that water is within 5 Å of any atom from D257 or D385. The Cα-Cα distance between L432 and K380 is measured to indicate the width of the gap between L432 and β2. The fraction of an active geometry and β3-strand formation are calculated by averaging the quantities measured every 1 ns in each 600 ns restraint-free simulations and every 50 ps in each 40 ns HREUS simulations.

The energy contribution of P1–P3' of β3 to the association event is calculated using the molecular mechanics Poisson-Boltzmann surface area (MMPBSA) method[75] with residue-wise energy decomposition. Since there is no lipid found at the interface between β3 and PS1, the binding event is considered in the water-solvated environment with dielectric constant $\varepsilon_{water} = 80$ and salt concentration of 0.15 M. Only the final 5 ns of each HREUS simulation is used to calculate the binding energy via the MMPBSA.py.MPI[75] program from the Ambertool18 package.

**APP substrate cleavage assay.** Mutant APP-based C100-His$_6$ substrates were generated by site-directed mutagenesis. Recombinant wild-type (WT) and mutant C100-His$_6$ substrates were expressed in *E. coli* BL21(DE3)$_{RIL}$ cells and purified by Ni-NTA affinity chromatography as described[34]. In brief, cells were grown for 4 h at 37 °C after induction with IPTG and lysed by sonication. Inclusion bodies containing C100-His$_6$ substrates were solubilized in 20 mM Tris (pH 8.5), 6 M urea,1 mM CaCl$_2$, 100 mM NaCl, 1% (w/v) SDS and 1% (v/v) Triton X-100 overnight at 4 °C and the tagged protein was bound to Ni-NTA beads. After washing of beads, bound protein was eluted with elution buffer (50 mM Tris (pH 8.5), 300 mM NaCl, 0.2% (w/v) SDS, 150 mM imidazole). Cell-free cleavage assays were performed with 0.5 μM of the respective substrates overnight at 37 °C as described[76] but using purified γ-secretase composed of PS1, NCT, PEN-2 and APH-1a[50] instead of detergent-solubilized HEK293 membranes as enzyme source. Generation of AICD was analyzed by immunoblotting with antibody Penta-His. The AICD species generated were analyzed by MADLI-TOF mass spectrometry (MS). To this end, AICD was immunoprecipitated in IP-MS buffer (10 mM Tris (pH 8.0), 140 mM NaCl, 0.1% *N*-octyl glucoside) with antibody 6687[4] and protein A-Sepharose. AICD was eluted from protein A-Sepharose with 0.1% trifluoroacetic acid in 50% acetonitrile, saturated with α-cyano-4-hydroxy cinnamic acid. MS analysis on a rapifleX MALDI Tissuetyper (Bruker) was performed as described[77,78].

**Statistics and reproducibility.** Data in the restraint-free simulations are generated from at least 4 trajectories (N = 2 or 4 for each PS1 protonation state) for each substrate length and each simulation contains 600 data points (time interval = 1 ns). Each PMF profile is generated using the WHAM algorithm with data collected from 20 to 60 ns with the reaction coordinate recorded every 2 ps in the HREUS simulations (Data at each exchange attempt are discard, in total 16,000 data points were collected in each replica). Error bars in the PMF profiles are shown as the standard deviation of the PMF curves calculated by taking the sampling time 20–22.5 ns, 20–25 ns, 20–27.5 ns, 20–30 ns, 20–32.5 ns, 20–35 ns, 20–37.5 ns, 20–40 ns, 20–42.5 ns, 20–45 ns, 20–47.5 ns, 20–50 ns, 20–52.5 ns, 20–55 ns, 20–57.5 ns, and 20–60 ns. Geometrical features including active site geometry, β3-strand formation, and water accessibility were calculated per 0.2 ns, in total 400 data points were collected in each replica. Data of each feature is binned according to the RC of the corresponding geometry. RC ranging from 7 Å to 18 Å are split into 11 bins with a 1 Å bin interval. Error bars are shown as the standard error of the data collected in each bin.

**Reporting summary**. Further information on research design is available in the Nature Portfolio Reporting Summary linked to this article.

## Data availability

All simulation data including input and output files and trajectory files are available at request from the authors. Source data, initial PDB files, and the code for generating the figures are available at https://doi.org/10.5281/zenodo.8000297. The uncropped blot of Fig. 4f is shown in supplementary Fig. S25.

## Code availability

Custom code for performing simulation and analysis is available from the cited references in the paper. In house analysis code is available upon request from the authors.

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

## Acknowledgements

We thank Dr. Manuel Hitzenberger for helpful discussions. The authors gratefully acknowledge computational resources provided by the Erlangen National High Performance Computing Center (NHR@FAU) of the Friedrich-Alexander-Universität Erlangen-Nürnberg. This work was supported by the Deutsche Forschungsgemeinschaft (DFG, German Research Foundation – 263531414/FOR2290; M.Z. and H.S.).

## Author contributions

S.-Y.C. performed MD simulation, analyzed the computational data, and wrote the manuscript. L.F. performed the cleavage assay, immunoblotting, and mass spectrometry experiments and wrote the manuscript. L.C.-G. provided the purified γ-secretase. H.S. and M.Z. supervised the project, provided resources, and wrote the manuscript.

## Funding

## Competing interests

The authors declare no competing interests.
