## [Peer Review File · Communications Biology]

Reviewers' comments:

Reviewer #1 (Remarks to the Author, also attached):

This manuscript describes a study of hybrid β -sheets formation of γ -secretase and its substrate APP and its implication on geometry and water access to its active site. The study employed a combination of computational and experimental techniques, including, MD simulations, APP substrate cleavage assay probed by immunoblotting and MALDI-TOF mass spectrometry. The manuscript is well-written, and most of the figures are clear.

The authors first used MD simulation to conclude the requirement of number of amino acids post scissile bond is 3 for the optimal active site geometry which is consistent with the previous studies on three pocket model by Bolduc et al. Then authors looked into hybrid β -sheet formation between γ -secretase and APP substrate with β sheet destabilizing M51P and L52P mutation and dissociation computationally and concluded that M51P and L52P mutation decrease γ -secretase cleavage efficiency due to increase in water accessibility because of mutation. This finding corroborates well with the experimental immunoblotting data.

Although these experiments are carefully executed here are some issues with the manuscript and would enhance the manuscript if authors take account into following suggestions.

1. Author mutated APP substrate β 3 residues (M51P and L52P) to destabilize the β sheet formation however similar mutagenesis to destabilize the β sheet must be performed in presenilin-1 to see whether we see the same effect as with the APP mutation.
2. The immunoblotting experiment to measure the activity of the γ -secretase is missing for the V50F and M51F mutant and would be helpful to include in the manuscript validating their computational result.
3. The MALDI-TOF spectra in the supplemental figure is not clear as the signal to noise ratio is very poor. I would suggest author to report the theoretical mass and experimental mass measured by MALDI-TOF for the AICD species, Also, I would suggest including the MALDI-TOF data for M51P which they missed in the manuscript

Reviewer #2 (Remarks to the Author):

This manuscript employs a combination of conventional molecular dynamics (MD) simulations, enhanced sampling, and experimental validation to investigate the role of the beta sheet formed between the amyloid precursor protein (APP) and γ -secretase in the process of substrate cleavage. Given the importance of γ -secretase as a drug target, understanding the substrate cleavage mechanism is crucial. Overall, the manuscript is well-written and presents interesting findings. However, there are a few comments that could further enhance the manuscript:

1. Figure 1E demonstrates that the fraction of the active state can reach >95% for APP. The initial conformation used in the simulation is 6IYC, which is an inactive state. It might be interesting to see when the conformation transition happened. However, the time-course of the conformation transition is not provided. Including this information in the Supplementary Information would be helpful.
2. The simulations conducted in this study are relatively short (600 ns for conventional MD and 60 ns for HREUS). Given the large system size, longer equilibration simulations may be necessary. While the author has assessed convergence for the HREUS simulations, it would be helpful to include a similar assessment for the conventional MD simulations.
3. The author employs the MMPBSA method to calculate the interaction between substrate and γ -secretase. However, this method works better for globular protein systems and may not be optimal for membrane proteins due to the differences in the environment. Using the recently developed MMPBSA

for membrane proteins (<https://doi.org/10.1021/acs.jcim.5b00341>) may be a more appropriate approach.

Garching, 27.4.2023

Dear Dr. Chong

Thank you for returning our manuscript entitled "Enzyme-substrate hybrid β -sheet controls geometry and water access to the γ -secretase active site" by Shu-Yu Chen, Lukas P. Feilen, Lucía Chávez-Gutiérrez, Harald Steiner and Martin Zacharias and the comments of the reviewers. In the following we like to comment on the concerns of the reviewers in a point-by-point response and indicate the changes and additions we have made to the manuscript. We include a version of the manuscript with all changes marked red.

Reviewer #1

This manuscript describes a study of hybrid β -sheets formation of γ -secretase and its substrate APP and its implication on geometry and water access to its active site. The study employed a combination of computational and experimental techniques, including, MD simulations, APP substrate cleavage assay probed by immunoblotting and MALDI-TOF mass spectrometry. The manuscript is well-written, and most of the figures are clear.

The authors first used MD simulation to conclude the requirement of number of amino acids post scissile bond is 3 for the optimal active site geometry which is consistent with the previous studies on three pocket model by Bolduc et al. Then authors looked into hybrid β -sheet formation between γ -secretase and APP substrate with β sheet destabilizing M51P and L52P mutation and dissociation computationally and concluded that M51P and L52P mutation decrease γ -secretase cleavage efficiency due to increase in water accessibility because of mutation. This finding corroborates well with the experimental immunoblotting data.

Although these experiments are carefully executed here are some issues with the manuscript and would enhance the manuscript if authors take account into following suggestions.

1. Authors mutated APP substrate β 3 residues (M51P and L52P) to destabilize the β sheet formation however similar mutagenesis to destabilize the β sheet must be performed in presenilin-1 to see whether we see the same effect as with the APP mutation.

Response: We thank the reviewer for the comment. Our paper was actually inspired by mutations in presenilin-1 that perturb or disrupt the possibility of β sheet formation with the substrate. This concerns the recent studies on the CryoEM structures of Notch-100 (Yang et. al., 2019, Nature), and APP-C83 (Zhou et. al., 2019, Science) that demonstrated the formation of a hybrid β sheet with the substrate. As a control the authors of these studies introduced mutations in presenilin-1 aiming to destabilize the hybrid β -sheet which resulted in a loss of enzymatic activity. Hence, the reviewer's suggestion has already been checked previously. Our complementary work demonstrates that APP substrate mutations that destabilize the E-S hybrid β -sheet also reduce the cleavage efficiency and define a minimum length that nicely explains why cleavage occurs in steps of 3 residues.

We extended a paragraph in the Introduction section (marked red, page 4) to emphasize that mutations in PS1 aimed at disrupting/perturbing the hybrid β -sheet formation with the substrate have already been performed and are consistent with the results of our study.

2. The immunoblotting experiment to measure the activity of the γ -secretase is missing for the V50F and M51F mutant and would be helpful to include in the manuscript validating their computational result.

Response: We thank the reviewer for the suggestion. However, exactly these 2 mutations have already been characterized experimentally very extensively in several published studies. This concerns the paper the reviewer mentioned (Bolduc et al., eLife, 2016) where these 2 mutations play a central role for the model presented by Bolduc et al (2016) which is consistent with the results of our study. However, in an independent study Bhattarai et. al. have reproduced the high AICD50/AICD51 ratio observed for the M51F mutation (Bhattarai et. al. 2020, ACS central science). Tan et. al. reported a very low A β 42/A β 40 ratio for the V50F mutation in the substrate and relatively high A β 42/A β 40 ratio for the M51F mutation (Tan et. al. 2008, J Neurochem). A similar result was also reported by Page et. al. (Page et. al. 2010, JBC). Hence, for the V50F and M51F mutations the published experimental evidence supports the results of our study and repeating these experiments is not necessary.

To further emphasize that the V50F and M51F mutations have already been studied extensively in agreement with our results we extended the paragraph on page 17 in our revised manuscript (marked red).

3. The MALDI-TOF spectra in the supplemental figure is not clear as the signal to noise ratio is very poor. I would suggest author to report the theoretical mass and experimental mass measured by MALDI-TOF for the AICD species, Also, I would suggest including the MALDI-TOF data for M51P which they missed in the manuscript.

Response: We agree with the comment of the reviewer. We have therefore exchanged the spectra for the WT with a new spectrum with a better signal-to-noise ratio in Figure S15. Due to the low cleavage of the L52P mutant we were not able to generate spectra with better signal-to-noise ratio for this mutant. Therefore, we now additionally provide the corresponding inhibitor controls to show that the peaks we detect arise from γ -secretase cleavage. As suggested by the reviewer, we now also include the theoretical masses and experimentally measured masses in Figure S15. Following the reviewer's suggestion, we also added the MALDI-TOF data for M51P. Since this mutant is virtually inactive, no peaks were detected. We also extended the statement on page 16 in the main text (marked red).

Reviewer #2 (Remarks to the Author):

This manuscript employs a combination of conventional molecular dynamics (MD) simulations, enhanced sampling, and experimental validation to investigate the role of the beta sheet formed between the amyloid precursor protein (APP) and γ -secretase in the process of substrate cleavage. Given the importance of γ -secretase as a drug target, understanding the substrate cleavage mechanism is crucial. Overall, the manuscript is well-written and presents interesting findings. However, there are a few comments that could further enhance the manuscript:

1. Figure 1E demonstrates that the fraction of the active state can reach >95% for APP. The initial conformation used in the simulation is 6IYC, which is an inactive state. It might be interesting to see when the conformation transition happened. However, the time-course of the conformation transition is not provided. Including this information in the Supplementary Information would be helpful.

Response: We have added a new Figure in the SI (Figure S9) to show the time-course of the active geometry, β 3-formation, and active site water numbers after *in silico* mutation of the active site residue. The catalytic hydrogen bond was formed quickly starting from the CryoEM structure (after *in silico* mutation of the active site residue) within the first 10ns in the simulations (essentially already in the

equilibrium phase of the MD simulation), which can be seen in the active geometry plot in the new Figure S9.

2. The simulations conducted in this study are relatively short (600 ns for conventional MD and 60 ns for HREUS). Given the large system size, longer equilibration simulations may be necessary. While the author has assessed convergence for the HREUS simulations, it would be helpful to include a similar assessment for the conventional MD simulations.

Response: As suggested by the reviewer in the previous question, we have added the time-course of recorded data in the SI (Figure S9) to check convergence. For the C-terminally shortened substrates (A β 50, A β 51) one can see consistently very quick loss of the hybrid β -sheet, accumulation of water at the active site and loss of active geometry. Intermediate behavior is seen for A β 53-bound and A β 52-bound γ -secretase cases with partial dissociation but infrequent re-association (for the A β 52 complex we performed 4 simulations to check the behavior). Exactly, because in continuous unrestraint MD simulations reversible dissociation and re-association is only infrequently observed motivated us to perform the HREUS simulations to induce dissociation/re-association and record the associated free energy changes. This was not sufficiently explained and we added a statement on page 9/10 (marked red) to emphasize the motivation for performing the HREUS free energy simulations.

3. The author employs the MMPBSA method to calculate the interaction between substrate and γ -secretase. However, this method works better for globular protein systems and may not be optimal for membrane proteins due to the differences in the environment. Using the recently developed MMPBSA for membrane proteins (<https://doi.org/10.1021/acs.jcim.5b00341>) may be a more appropriate approach.

Response: In the MMPBSA method for membranes indicated by the reviewer the protein is embedded in dielectric medium (with low dielectric constant) representing the membrane with a preset width of membrane thickness. In our case the membrane thickness is around 37 Å (with upper slab at +18.5 Å and lower slab at -18.5 Å). However, in our system the important β 3-strand and hybrid- β -sheet region is solvated and not in contact with lipids of the membrane region (in fact the lipids are relatively far away from this region (> 15 Å)). In fact, the protein region at the focus of our study is pointing toward the aqueous intracellular region forming a dielectric boundary directly to water and not to the membrane. It is therefore more realistic to treat this system using the conventional MMGBSA approach. Future developments that allow for irregularities and interruptions of the implicit membrane parts might be useful and we plan to investigate this in future studies. We added a statement concerning the use of the standard MMPBSA approach on page 13.

Of note, we want to mention that we tested such MMPBSA method for membranes on a related task in a previous paper (Hitzenberger and Zacharias, 2019 ACS Chemical Neuroscience). We found that moving an APP substrate with a Lys at position 28 (K28) (and using an MPBSA for membrane approach) involves an unrealistically high calculated electrostatic penalty ($\Delta\Delta G$ of ~ 30 kcal/mol) which if taken as real essentially prevents formation of shortened A β -products (because the high penalty would prevent any movement of K28 into the “membrane regime”). The K28 remains partially solvated in explicit solvent simulations and placing the implicit periodic membrane regardless of the actual water distribution can cause strong overestimation of the penalty to move the substrate towards the enzyme active site because the actual dielectric constant of the solvated part is replaced by the implicit membrane ($\epsilon_{\text{membrane}} \approx 2$). Hence, also here the current simple membrane representation in the special MMPBSA membrane approach is insufficient to realistically represent the actual water distribution.

Finally, we like to thank the reviewers and you for the comments that, we believe, improved our manuscript and hope that with the additions and changes we have made to the manuscript it is now acceptable for publication in *Communications Biology*.

Yours sincerely,

Martin Zacharias

REVIEWERS' COMMENTS:

Reviewer #1 (Remarks to the Author):

The revision the authors provided is sufficient and addressed my comments appropriately.

Reviewer #2 (Remarks to the Author):

The authors have addressed all my comments.